# Fully Self-Supervised Class Awareness in Dense Object Descriptors

**Denis Hadjivelichkov**
Department of Computer Science
University College London
United Kingdom
dennis.hadjivelichkov@ucl.ac.uk

**Dimitrios Kanoulas**
Department of Computer Science
University College London
United Kingdom
d.kanoulas@ucl.ac.uk

**Abstract:** We address the problem of inferring self-supervised dense semantic correspondences between objects in multi-object scenes. The method introduces learning of class-aware dense object descriptors by providing either unsupervised discrete labels or confidence in object similarities. We quantitatively and qualitatively show that the introduced method outperforms previous techniques with more robust pixel-to-pixel matches. An example robotic application is also shown - grasping of objects in clutter based on corresponding points.

**Keywords:** Self-Supervision, Descriptor Learning, Object Correspondence

## 1 Introduction

Self-supervised methods allow models to train with no manual labelling, thus improving the ability to scale and learn from more data. In robotic applications, they could potentially allow a robot to learn continuously 'in the wild'. Robots of the future would benefit from the ability to learn how to interact with many objects, including unseen novel ones. One particular problem is finding accurate pixel correspondences between similar objects in such a way that a robot agent recognises the corresponding parts between two items and manipulates them in similar ways.

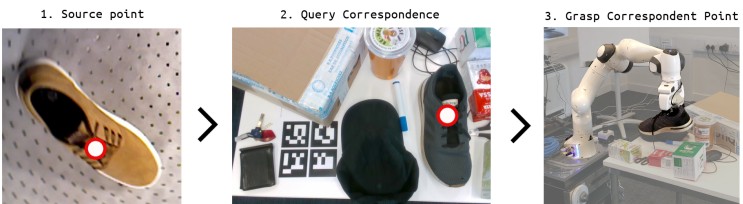

Figure 1: Correspondence (red circle) produced by our *DON+Soft* method used for robot grasping.

A recent work producing self-supervised object descriptors, namely Dense Object Nets [1] (DONs), has been applied as a representation learner for visuomotor policies [2, 3] improving onto previous end-to-end methods [4]. While DONs produce excellent point correspondences between similar objects, without access to ground truth classes they fail to find robust matches in multi-object scenes and produce false positives. In this work, we propose a method for class-aware Dense Object Network and show that it leads to more informative descriptors that are better suited for correspondence generation in multi-object scenes. We present two variants exploring different ways of representing the object classes: "hard" discrete labels and "soft" labels, reflecting our confidence in the similarity between the objects. We demonstrate that our method is better than the state-of-the-art in finding correspondences in multi-object images, while simultaneously improving performance of on single-object images. A real example application is shown using a mobile manipulator[1]. This method could enable future work on more complex tasks such as unsupervised learning of category-level affordances and manipulation of objects in clutter.

---

[1]See https://sites.google.com/view/multi-object-dense-descriptors for video.

5th Conference on Robot Learning (CoRL 2021), London, UK.

## 2 Related Work

In this section we briefly overview recent advances in descriptor learning. We then review methods for unsupervised classification and class disentanglement. Our work focuses on self-supervised learning of descriptor representations, such that the descriptors of similar points belonging to similar object are consistent across time, view, and object instance. Our method builds upon Dense Object Nets (DONs) [1] which makes use of 2D pixel to 3D point projections as a supervisory signal. The method is described in more detail in Sec. 3. Most other works on descriptor learning such as [5, 6, 7] make use of fully- or weakly-supervised learning requiring large amounts of human-annotated data and do not make efficient use of depth data, which robots usually have access to. Several methods relieve the need for annotations by using 3D CAD models of the observed objects [8, 9], however this is not applicable when encountering novel objects. Many works make use of spatio-temporal consistency across video frames, using tracking trajectories or multiple modalities: [10] learns unique descriptors via optical flow contrasive learning from dynamic data. Grasp2Vec [11] learn similar features through autonomous robot intereaction. The semi-supervised S3K [12] uses multi-view consistency to produce keypoints with high accuracy. With only few ground truth examples, it is learns 3D keypoints by estimating the projections of 2D keypoints. It proves to be useful for applications such as cable plug-in, where keypoints do not have to be bound to a physical location on the object, as long as they are consistent across views. However, it does not produce dense correspondences. We are not aware of any work that produces *pixel-pixel* correspondences across or between *multi-object* scenes, based on unsupervised or self-supervised training without ground truth models or per-image labels, which is introduced in here.

For our method to work, we require a way of assessing the similarity or dissimilarity between objects and disentangling their representations. This can be done by assigning some discrete class labels to each object or measuring our confidence in the similarity of object pairs. Most recent works use unsupervised clustering techniques such as mean-shift and K-Means in combination with embeddings [13, 14, 15]. Patten et al. [16] improve on previous approaches by considering geometric similarity and experience queries. Object-Contrastive Network [17] finds object candidates via Faster-RCNN, embeds them and then samples triplets based on nearest neighbours in embedding space. Recently, Tan et al. [18] showed that clustering of object embeddings alone is not enough. Instead, they propose a method for similarity measurement and triplet sampling via random walker graphs. Their method demonstrates state-of-the-art results on various benchmarks. We use it as an inspiration to our method and provide further details in Sec. 4.2.1.

## 3 Dense Object Nets

Dense Object Nets [1] which are largely based on Schmidt et al. [19] are the foundation of our method. In this section we explain how they work and identify their current limitations. DONs learn a dense descriptor representation of RGB inputs that is applicable to both rigid and non-rigid objects. It uses the strong prior of depth information taken from an RGB-D camera, and knowledge of a robot manipulator's pose to project 2D image pixels onto a reconstructed 3D model, thus learning correspondences between different views of the same 3D points. In particular, this is done by having a robot point a camera towards an object and move around to capture it from different views. In Siamese fashion, the model is fed pairs of augmented images from which pixels are randomly sampled. Augmentations consist of random background, color jitter, crops, and scale. Pixels corresponding to the same 3D point are considered 'matches' while any other random pair of pixels are considered 'non-matches'. The loss is the sum of match and non-match losses defined as follows:

$$L_{match} = \frac{1}{N_+}\sum_{N_+}||d_a - d_b||_2^2 \tag{1}$$

$$L_{non-match} = \frac{1}{N_m}\sum_{N_-}\max(0, M - ||d_a - d_b||_2)^2 \tag{2}$$

where $d_a$ and $d_b$ are the output descriptors at the given pixel pair and $N_+$ is the total number of match pairs. Similarly, $N_-$ is the total number of non-matches. $M$ is a margin hyperparameter determining the minimum distance between non-match pairs - if a non-match pair is closer than $M$, the loss is increased. $N_m$ is the number of these matches that is below the margin. While $L_{match}$ encourages

true pairs to be similar, $L_{non-match}$ ensures the pixels don't all converge to the same descriptor. Since the robot has no knowledge of the object's class, the learned descriptors tend to share the same descriptor space for all objects. While the original work shows that when the ground truth class is known, the descriptor subsets can be distinctly separated, it does not propose how to implement this and remove the use of ground truth.

## 4 Method

The method presented in this section tries to solve the problem of disentangling the descriptor spaces shared by DON descriptors by including class awareness.

### 4.1 Problem Setup

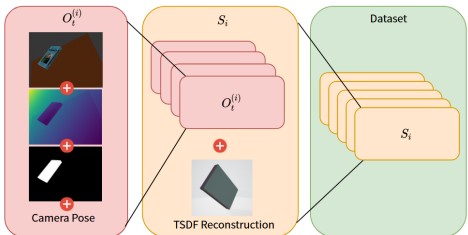

Figure 2: Data Organization: each data-point consist of an RGB, Depth, and Mask image and a camera pose. Sequences are comprised of data-points, while the dataset is composed of sequences. A 3D TSDF reconstruction is automatically generated by combining the RGB-D views of a sequence during data collection.

The inputs used for training are a set of $n$ sequences $S = \{S_1, S_2, ..., S_n\}$, where each sequence contains timed data for a single object $S_i = \{O_1^{(i)}, O_2^{(i)}, ..., O_t^{(i)}\}$. The data $O_t^{(i)}$ for timepoint $t$ and object $(i)$, consists of a camera pose, full RGB and Depth image and object mask. A 3D TSDF reconstruction is also generated for the whole sequence. See Fig. 2. In this paper, each *training* sequence strictly contains a single instance of a single object, while during *inference* we can work with both single and multi-object sequences. We discuss how the method could be adapted to train from multi-object sequences in Sec. 6. We refer to each distinct object as an **instance class** and its grouping as a **category**, e.g. different hats would belong to different instance classes but the same category. Given $S$ we want to learn visual descriptors that are robust over different objects.

### 4.2 Class-Aware Dense Object Nets

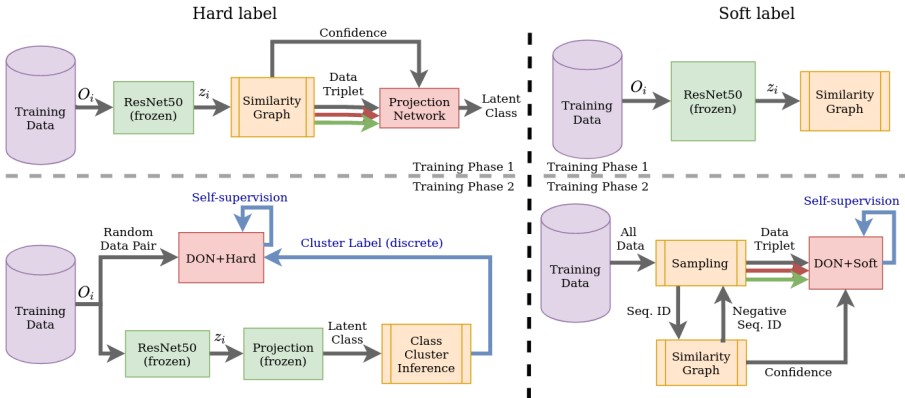

Figure 3: The simplified data flow for the hard (left) and soft (right) label variants of our method. Blue arrows signify self-supervisory signals, grey - the main data flow, green and red arrows denote positive and negative triplet pairings. (best viewed in colour)

To disentangle the descriptor subsets used by different objects, most straightforward is to find nonmatch pairs between them and learn to repel them. We considered two variants: (i) *DON+Hard* producing discrete "hard" labels by assigning each instance to a cluster or (ii) *DON+Soft* using similarity-based triplet sampling and continuous "soft" confidence scores. Both variants use the same DON model for dense correspondence, and similarity graph for object disentanglement, differing mainly in data flow, sampling and loss generation, as described below (see Fig. 3).

### 4.2.1 Similarity Graph

Since we assume access object sequences but not their class labels, the method requires a way of understanding each object's identity or class grouping in a different way. Here we propose object disentanglement based on a recent work [18] which consists of a similarity graph (SG) and random walker sampling (RWS). SG is an undirected graph with each node representing a training sequence and each edge the similarity between the node pair. Thus, given any node in the SG, we can find which other nodes are most probably similar to it. RWS is the sampling of node pairs by "walking" on edges with transitional probability proportional to the edge weights.

The similarity graph is undirected, with each training sequence representing a node $S_i$. The edges between each pair of nodes is determined by *similarity weights* $W(S_i, S_j)$ which are calculated using a similarity and dissimilarity measure, $W^+$ and $W^-$, respectively. Each RGB frame from each sequence is pre-processed via a pre-trained ResNet before being used for edge weight calculation. Other inputs were also considered in Sec. 5.

The frames of each sequence are clustered into $N_l$ *local intra-sequence* groupings via K-Means. Since the sequence contains only a single object, these clusters tend to contain different view points of that object. This solves the over-representation issue that some sequences may suffer from, e.g. if one the views is prevalent over others. Each sequence $S_i$ then has $l$ corresponding cluster centroids $C_{l,i}$. The centroids have the same dimensions as the input features. Through minimum linear sum assignment [20] with assignment matrix $X$, the minimal cluster distance sum is found and used as a dissimilarity measure (See Eq. 3). In our version of this weight, it is normalized by the number of pairs $N_p$ and the number of dimensions $N_d$.

$$W^-(S_i, S_j) = \frac{\min_X \sum_k \sum_m X_{k,m} ||C_{k,i} - C_{m,j}||_2}{N_p . N_d} \tag{3}$$

s.t. Each cluster being assigned to at most 1 pair

Similarly, all frames from all sequences are clustered together in $N_g$ *global inter-sequence clusters* via K-Means. By using a large $N_g$, the frequency at which two objects share the same clusters is indicative of how similar those objects are, shaping the similarity measure $W^+$, as in [18].

Finally, the two measures are combined along with a tuning parameter $\lambda$ which determines the importance of each measure over the other (see Eq. 4). The final weight is strictly non-negative.

$$W(S_i, S_j) = \max\left(\lambda W^+(S_i, S_j) - W^-(S_i, S_j), 0\right) \tag{4}$$

Given a starting node in the graph, a transition probability distribution $p_t$ is formed proportional to the connected edge weights. By sampling a transition from $p_t$, a 'random walk' is done to a node that is more likely similar. Similarly, sampling from $(1 - p_t)$ is used to reach more probably dissimilar nodes. We propose a dissimilarity confidence $c$ in the pairing equal to the min-max normalized $W^-$. This is a replacement of the confidence proposed in [18] which was found to be very sensitive to $\lambda$.

### 4.2.2 Dense Object Net

For DONs we use a 34 layer, 8-stride ResNet architecture. Its output descriptor image is upsampled to the original input size and has shape $(H, W, D_{desc})$, where $H$ and $W$ are the input image height and width, and $D_{desc}$ is the number of descriptor dimensions.

**Hard Classification:** As in [18], for *DON+Hard* the triplets (anchor $A$, positive $P$, and negative $N$) of sample frames are generated using the similarity graph and fed into a projection network comprised of two fully connected layers with ReLU activations. The projection network is trained via triplet loss scaled by our confidence $c$ and margin $M$. The loss is made non-negative as shown:

$$L = ReLU\left(||A - P||_2 - ||A - N||_2 + c.M\right) \tag{5}$$

The resulting projection output is a continuous latent class representation. Finally, the representations of the training data are grouped into $K$ clusters, determining the discrete classes. In inference time, assuming the input belongs to one of the existing classes, it can be projected and matched to the closest cluster. We denote this as training phase one. In the second phase, the DON iteratively

accepts two kinds of image pairs as inputs with equal probability: (i) same sequence pairs that consist of two different frames of the same object from the same sequence and (ii) different sequence pairs that consist of two frames from random different sequences. In the different sequence scenario, the objects' discrete classes are inferred by assigning them to one of the $K$ clusters - if their classes are the same, the pair is ignored, otherwise - non-match pixel pairs are generated and used for DON loss. In the same sequence, both match and non-match pairs are generated.

**Soft Classification:** For the DON+Soft variant, the similarity graph is generated with all available sequences (training phase one). A random sequence is sampled and a negative sequence pairing is randomly chosen based on the transition probabilities of the similarity graph. Two frame samples are taken from the initial sequence (anchor $A$ and positive $P$) and one from the negative sequence, $N$, forming a triplet. The triplet is passed through the DON. Only non-matches are generated for the negative pair. A combination of the DON pixelwise match and non-match losses is used (See definitions in Eq. 1 and Eq. 2). The negative pair does not have any matches, instead its $L_{non-match}$ is scaled by the confidence $c$ in the negative pairing, thus if there is strong belief that the pair is from the same class, the loss is smaller. The total loss is defined as follows:

$$L = L_{pos,non-match} + L_{pos,match} + c.L_{neg,non-match} \qquad (6)$$

## 5  Experiments

With our experiments we want to answer how effective our method is at separating object descriptors based on object categories. In this section, we investigate this both qualitatively and qualitatively.

### 5.1  Experimental Setup

***Training Data:*** For training we use (i) real data from [1] and (ii) newly produced simulated data (see Fig. 4). For the simulated data we placed a Gazebo-simulated Franka Emika Panda manipulator equipped with a RealSense RS200 RGB-D camera, and captured objects from the 3D GEM dataset [21] in randomised poses. A summary of data is presented in Table 1. Samples of the datasets are shown in Fig. 4. Note, that because of the use of background augmentation in DON, the unrealistic scene in the simulation does not affect training.

| Name | Scenes | Images | Categories | Instance Classes |
|---|---|---|---|---|
| Real Shoes and Hats (S&H) | 53 | 25754 | 2 (shoes and hats) | 8 |
| Real Robots (Bots) | 10 | 6143 | 3 (3x robot toys) | 3 |
| Simulated Cans and Books (C&B) | 30 | 22155 | 2 (books and cans) | 8 |

Table 1: Summary of data

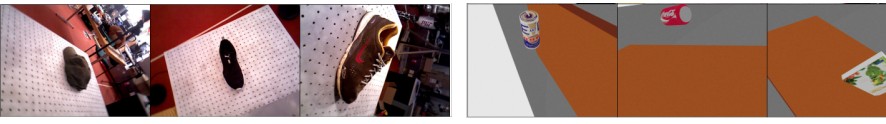

Figure 4: Samples of RGB object images used as training data, real (left) and simulated (right).

***Hyperparameters:*** Unless otherwise specified, all trainings are done for $3,500$ iterations with $D_{desc} = 5$ descriptor dimensions. The learning rate is $10^{-4}$, with a decay of 0.9. DON margin parameter $M$ is set to 0.5. For the similarity graph, we used $N_l = 5$ and $N_g = 300$ clusters, and $\lambda = 0.1$. For *DON+Hard*, the projection network is trained for 500 steps (phase one) before proceeding.

### 5.2  Object Disentanglement Experiments

***Setup:*** For our method to work, object representations need to be effectively disentangled. We test the ability of the full classification model (i.e., consisting of the Similarity Graph and Projection Network, as in *DON+Hard*) to work with DON-compatible object sequences. We consider three input types: (i) raw RGB; (ii) output of pre-trained frozen ResNet50; and (iii) masked Descriptor Image produced by a pre-trained DON. All inputs have been cropped to fit only the object and interpolated to a fixed size. We test both on simulated (C&B) and real (H&S) objects, where each object category contains 4 separate instance classes. We take 600 random images for training of the

model and then test its cluster assignment on $2,000$ random test images of known object instances from unseen sequences. Inference into clusters is done via K-Means with $K = 2$ and $K = 8$ clusters corresponding to the number of object categories and instance classes respectively. Note that $K$ does not affect training. Cluster assignment to ground truth classes is done by minimum linear sum assignment [20]. The model accuracy is then the fraction of images that are correctly assigned.

We also aim to show whether the descriptors produced by our methods are effective in separating the classes and how they compare to DONs trained with ground truth labels and without any labels. The models are trained on C&B. The descriptors of 50 random images are produced and 400 on-object pixels are sampled from each. These pixels are then projected via TSNE. For *DON+Hard*, $K$ is set to 5 class clusters to highlight the difficulty of not knowing the exact number of classes.

***Results:*** As can be seen in Table 2, the ResNet features appear far superior over other inputs. Not only are they effective in separating the features into categories but also into object instance classes.

Moreover, as it can be seen in Fig. 5, even without any parameter informing the *DON+Hard* model that there are two object categories, we observe a clear separation of two clusters containing only objects from that category. Our qualitative evaluation demonstrates that the method is effective in separating objects, having similar objects share similar descriptor spaces. Comparing it with the *DON+Soft*, there is no clear discrete separation between the classes, but there's a gradual transition between them. We want to note that this is closer to how we interpret objects ourselves - if two objects have some similar features such as a colour, a handle, or other, we notice that similarity, while still detecting the overall difference between the objects.

| Data | Raw Image | ResNet Features | Masked Descriptor |
|---|---|---|---|
| C&B @$K = 2$ | 52.93% | **99.20%** | 96.31% |
| C&B @$K = 8$ | 22.13% | **95.25%** | 76.23% |
| H&S @$K = 2$ | 51.00% | **97.80%** | 85.26% |
| H&S @$K = 8$ | 16.07% | **93.12%** | 71.09% |

Table 2: Comparison of classification accuracy for different input types.

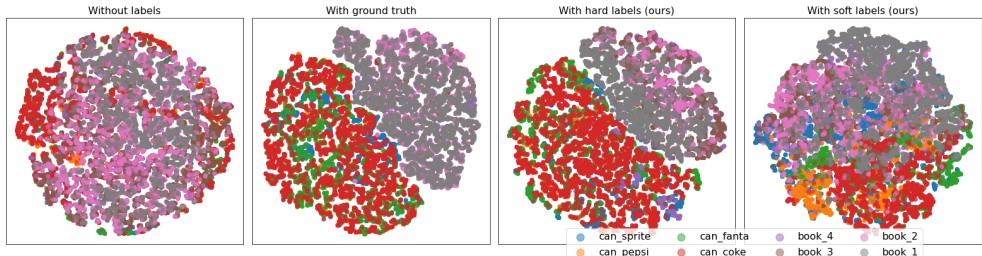

Figure 5: TSNE projection of randomly picked pixels from descriptor images produced by DONs trained without classification, with discrete ground truth labels, and our *DON+Hard* and *DON+Soft* methods. Eight classes are used from two categories (books and cans).

## 5.3 Descriptor Match Accuracy Experiments

***Setup:*** Given the descriptor at a random point from a query image, its best match in a target descriptor image is found as the one with minimal $L2$-norm. When working in single object scenarios, the original DON finds good semantic correspondences between semantically corresponding objects. However, as soon as there are multiple objects in the target image, the quality of the matches falls. To test this, we create new images, with unseen object instances in visually cluttered scenes, either through image collages or **real** photos (see Fig. 6). We manually label points on a set of 40 single-object images out of *H&S* (not used for training) and our 20 multi-object images, each having labelled semantically significant locations (when not occluded). For example, this includes distinctive points such as tip, top and back of a hat. This results in 432 pixel pairs for multi-object matches and 1218 for single-object matches. The closest descriptors of the labelled points in the single-object images are found in query single- or multi-object images. The distance between the best match and the labelled point is the error. We plot the normalised error versus the fraction of

images as a cumulative distribution (CDF). Comparison is done between our models and DON. As upper baseline - models trained with ground truth DON+GT (as per [1]) and DON+Soft+GT, while DAISY [22] provide a non-semantic lower baseline. All trained on H&S. A DON+Soft trained on three extra classes (on H&S+Bots) is also compared with DON+Soft.

The quality of matches is evaluated by picking random points on an image and comparing the correspondences on a test image with novel instances of known classes. Models trained on H&S.

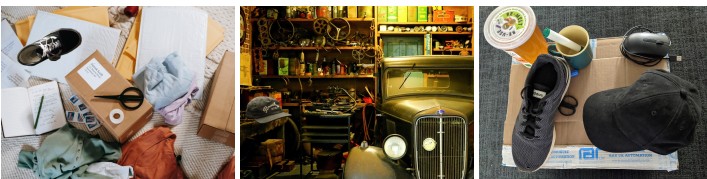

Figure 6: Examples of multi-object test images in cluttered scenes: shoe, hat, and both.

***Results:*** The CDFs is shown in Fig. 7. It can be seen that the change in performance with increase of classes is minimal, while there is an even smaller error in single object matches. As it can also be seen in Fig. 8, *DON+Soft* outperforms the other methods.

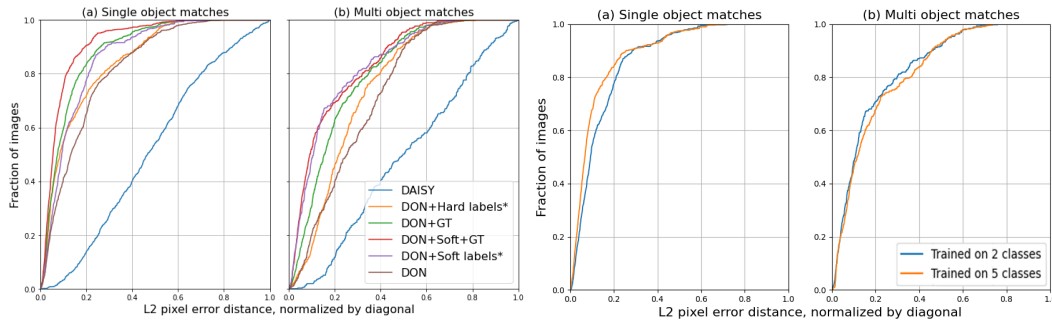

Figure 7: CDF of L2 pixel error distance normalised by the image diagonal for single and multi-object queries. **Left:** comparing our methods (denoted by *) with DON. *DON+Soft* labels (purple) most effectively retain good match correspondences in multi-object queries with undeterred performance for 67% of matches. **Right:** comparing *DON+Soft* performance when trained on shoes and hats only vs. trained with three extra classes. (best viewed in colour)

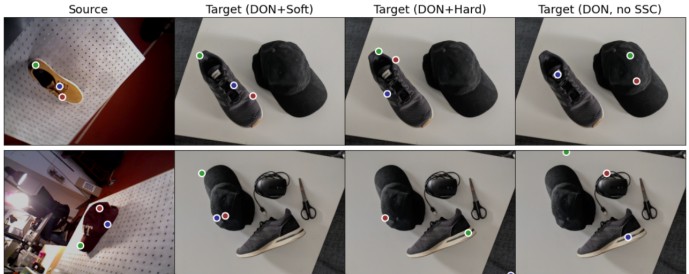

Figure 8: Random points from query images (left column) matched to the closest descriptor points in the target images with *DON+Soft*, *DON+Hard*, and no self-supervised classification (original).

***Discussion:*** As it can be seen both qualitatively and quantitatively, DON+Soft produces more robust correspondences in multi-object scenarios. Surprisingly there was a slight improvement in single-object matches. We believe this could be due to the contrast between objects encouraging the descriptors to be more specific and informative.

## 5.4 Real Robot Experiment

***Setup:*** To showcase the applicability of the model on a real robot we set up the following experiment. We use the mobile manipulator MPPL's [23] Franka Emika Panda arm equipped with a RealSense D435i RGB-D camera. Firstly, a point is selected on an image of an object from the

training set, in this case, a shoe or a hat. Then the robot finds the best match within its environment, by querying a DON+Soft trained on H&S. The correspondent pixel is projected into 3D space via the depth image and camera-to-world transformation. Finally, the robot produces a grasp for said point. Note that the objects used in the experiment are not part of the training set, and the camera used by the robot is different from the ones used for the data gathering. The clutter observed by the robot contains both known objects on which the model is trained and novel ones - to check that the model does not confuse the correspondences with either. The source object, source point on object as well as the clutter configurations were varied to ensure robustness. A total of 20 trials were done.

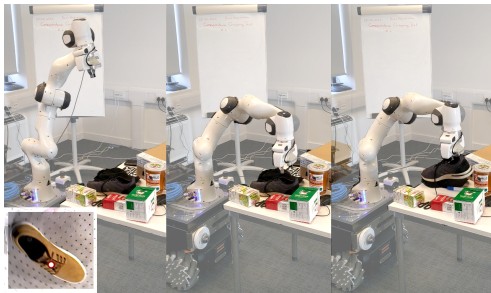

Figure 9: Grasping given a point: the model finds the best match to a selected point (shown in corner) within the robot's view, the pixel is projected into 3D space, the robot produces a grasps.

***Results:*** See Fig. 9. The correspondences are correctly placed on the queried object during all trials. One failure occurred when the point found was at the edge of the object - it couldn't be picked.

## 6  Conclusions

**Limitations:** Our method shows more robust matches than its predecessor in multi-object scenarios. One failure case we observed is how our method deals with occlusion. Matches to occluded points are found in completely wrong locations which indicates that although the descriptor subsets for similar object are similar, the individual distances between descriptors within the subsets are big. Querying for best-matches does not consider neighbouring descriptors, geometric consistency or other metrics which likely causes some of the other errors in our best-match correspondences. The DON in its current form is unable to work with dynamic training scenes. This could potentially be alleviated with the use of multiple viewpoints or tracking pixels with optical flow. Moreover, since the current system relies on masks of the input sequence objects, it could be extended to multi-object training scenes by adding a semantic segmentation network. The use of K-Means in *DON+Hard* leads to computational complexity scaling with number of sequences, as well as dealing with a constant number of clusters $K$ when the real number of classes is unknown. In our exploratory work, to alleviate these issues we tried Mean Shift Clustering, Bayesian Gaussian Mixture Models and DBSCAN [24]. However, it was found that the classification accuracy on training data dropped significantly, likely because the latent classes are not described by normal distributions. *DON+Soft* would be unaffected by this since it purely relies on similarity measures.

**Applications:** This work enables applications that might treat different classes of objects differently. Here we provide three pick-and-place examples: (i) given some knowledge base of different objects and graspable points on them, the method can be used to automatically detect the closest match points even in cluttered multi-object scenes; (ii) or the opposite way, the model could see an object, infer its latent class and match it to the closest of its class clusters, thus generate the best grasping points for that class; (iii) alternatively, the model could be used to observe how people interact with objects, and learn which descriptor-points are suitable grasping points for each self-supervised class. In terms of more complex applications, we believe this representation could pave the way toward methods such as self-supervised robot affordance learning and action imitation.

**Future Work:** In this paper, we extended an existing method for self-supervised object descriptors to perform more accurately in multi-object queries and cluttered scenes. We showcase its improvement over its predecessor both qualitatively and quantitatively. Further analysis of the method could explore how having novel objects vs known objects in the background of a scene affects the performance, i.e. would it be better for the model to be trained on everything it sees, or does performance degrade once an object has been 'learned'. Future work will be focused on making the descriptors more useful by grounding them to actionable representations, and extending the training toward training in the wild from dynamic unstructured scenes.

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
