# OpenReview forum: "Fully Self-Supervised Class Awareness in Dense Object Descriptors"
_robot-learning.org/CoRL/2021/Conference — CoRL2021 Poster_

### Official Review · Reviewer_RTPV · 2021-07-17

**Originality:** Very Good
**Technical Quality:** Very Good
**Clarity Of Presentation:** Very Good
**Impact:** 4

**Recommendation:**

Strong Accept: I recommend accepting the paper and will argue for my recommendation even if other reviewers hold a different opinion.

**Summary:**

This paper resolves a limitation of Dense Object Nets in scaling to multiple objects in a self-supervised fashion. The authors introduce a fully self-supervised module which does need the label of which object is in the camera view.  Specifically, in the original Dense Object Nets, in the multi-different-object case, a small amount of human supervision was needed in order to apply "cross-object loss" as stated in the original paper.  With this limitation resolved, this in theory aids the ability of Dense Object Nets to potentially larger scale than before, since the method is more fully self-supervised.

**Issues:**

This is a list referring to things mentioned previously:
1. Paper's key contribution could be more precisely stated.  For inspiration please see my Summary.
2. I would recommend the paper title be slightly revised to "Fully Self-Supervised Class Awareness in Dense Object Descriptors". Please see discussion above.
3. Please also cite Schmidt et al (see above)
4. Please see comment above regarding comment on S3K.

**Reviewer Expertise:**

Excellent: Expert knowledge on the topic of the paper

**Strengths And Weaknesses:**

I think this is good work.  That said, there are a few areas for improvement.

Strengths:
- The contribution is clean. (Although it could use some revision in making it more clearly stated. Please see my Summary for an example of how the main contribution could be more specifically stated.)
- Figure 3 is very good at explaining the main idea.
- Figure 6 represents a substantial improvement relative to the original Dense Object Nets paper, particularly for multi object matches.
- The real robot experiment does not really add that much to the core academic argument, but still, it is appreciated.
- Figure 4 shows compelling clustering.

Weaknesses
- Section 4.3.2, which is maybe the most critical section, is not very clear.  How is each C computed?  This section should be improved for clarity if the paper is accepted.  Please just run it by multiple rounds of somebody who is not familiar with the work, and see if they can read it and understand exactly all of the symbols and how they are computed.
- The cost of labeling which object is present in each scene, as was assumed in the original Dense Object Nets paper, is not actually that large of a cost.  Yet still, it is nonzero cost.  The missed opportunity though is in showing that, without this small cost, maybe this method can be scaled to a much larger scale than before. Especially since the authors set up a simulated pipeline, maybe they could have scaled this to 1000s of objects.  This would be interesting but wasn't done.

I would recommend the paper title be slightly revised to "Fully Self-Supervised Class Awareness in Dense Object Descriptors".  The current title, "Self-Supervised Dense Object Descriptors with Class Awareness" is not as precise, since in the previous Dense Object Nets paper, there was the ability to do distinct multi-object descriptors.  The difference though is that in order to do this before, the images needed to be labeled with which object they corresponded to.  "Fully Self-Supervised Class Awareness in Dense Object Descriptors" is accordingly more precise.

Also a related work comment is incorrect.  It says: "We are not aware of any work that produces pixel-pixel correspondences in multi-object scenes, based on unsupervised or self-supervised training without ground truth models, which is introduced in this paper."  This should be revised to include a couple key words, called out with asterisks. "We are not aware of any work that produces pixel-pixel correspondences *across or between different* multi-object scenes, based on unsupervised or self-supervised training *without per-image labels of which object is present*."  Note that many SfM methods would be able to find dense correspondences in a multi-object scene if all the images are from the same scene, so the *across or between different* scenes is critical.  Also, the original Dense Object Nets work required a label of which object is present in order to scale to multiple distinct objects.

Please also cite Schmidt et al, "Self-supervised visual descriptor learning for dense correspondence", which Dense Object Nets is heavily based on, but is missing from the current citations.

Minor note:
- The related work, referring to S3K, says: "it does not produce correspondences".  But keypoints are correspondences.  They are just sparse correspondences, instead of dense.  So this statement should be revised to say that S3K does not produce dense correspondences.

**Summary Of Recommendation:**

I think this paper provides a clean contribution.  The experiments are sufficient to show the method works.  And getting rid of the "which object is present" label needed from DON before could unlock additional scaling of the method.

I put "Strong Accept" because I am willing to argue for this paper's acceptance.  That said, it would be a valid view that the contribution is incremental and of a small enough increment that it may not need to be accepted.  I'm willing to argue the contrary based on any specifics that other reviews point out, but I could ultimately support either decision for this paper.

---

> ### Author Response · Authors · 2021-08-23
> **Comment Answer to Reviewer RTPV**
>
> **Summary of major changes:**
> - Changed title to "Fully Self-Supervised Class Awareness in Dense Object Descriptors''
> - Major rewording of Section 4.2.1 Similarity Graphs to be more clear and informative.
> - Additional figure showing the structure of the data is added in Section 4.1. Problem Setup.
> - Experimental section now explicitly separates the setup and results of each experiment. Data is summarized in a table and experiments state which data they use.
> - The system figure (Fig. 3 in revision) has been changed to include all outputs and be more clear, rather than abstract.
> - Some references have been added to related work per suggestion of the reviewers - (Pirk et al., Schmidt e al., Grasp2Vec)
> - Experiments with a lower and upper baseline were ran for the quantitative evaluation. These have been shown in large scale in the supplementary materials.
> - To fit these changes in the paper's 8-page limit, some sections were re-ordered, trimmed and changed. Further figures answering reviewers concerns have been added to the supplementary files.
>
>
>
> Dear Reviewer, we would like to thank you for the review and the suggestions. Below, we have addressed your comments.
>
> >    Section 4.3.2, which is maybe the most critical section, is not very clear. How is each C computed? This section should be improved for clarity if the paper is accepted. Please just run it by multiple rounds of somebody who is not familiar with the work, and see if they can read it and understand exactly all of the symbols and how they are computed.
>
> Thank you for the advice, we have reworded the section for clarity (as much as we could given the 8-pages constraint). Please see Sec. 4.2.1. in revised paper (where we also describe how C_ij are the K-Means centroids).
>
> > The cost of labeling which object is present in each scene, as was assumed in the original Dense Object Nets paper, is not actually that large of a cost. Yet still, it is nonzero cost. The missed opportunity though is in showing that, without this small cost, maybe this method can be scaled to a much larger scale than before. Especially since the authors set up a simulated pipeline, maybe they could have scaled this to 1000s of objects. This would be interesting but wasn't done.
>
> Actually, we found that with our current pipeline it was difficult to find suitable models that are fully usable in the Gazebo simulator and that have multiple different instances belonging to the same class.  We had to discard many of the models present in the 3D GEM dataset as, for example, some just fell apart. We are planning on moving to a different simulator (e.g. a photo-realistic one, such as NVIDIA Isaac) for a more easy integration with other 3D models, and possibly a much better sim2real transfer.
>
> >I would recommend the paper title be slightly revised to "Fully Self-Supervised Class Awareness in Dense Object Descriptors".
>
> > Also a related work comment is incorrect. It says: "We are not aware of any work that produces pixel-pixel correspondences in multi-object scenes, based on unsupervised or self-supervised training without ground truth models, which is introduced in this paper." This should be revised to include a couple key words, called out with asterisks. "We are not aware of any work that produces pixel-pixel correspondences across or between different multi-object scenes, based on unsupervised or self-supervised training without per-image labels of which object is present."
>
> >Please also cite Schmidt et al, "Self-supervised visual descriptor learning for dense correspondence"
>
> > This statement should be revised to say that S3K does not produce dense correspondences.
>
> The suggested changes have all been included in the new revision.

---

### Official Review · Reviewer_ZCDs · 2021-07-18

**Originality:** Fair
**Technical Quality:** Fair
**Clarity Of Presentation:** Fair
**Impact:** 3

**Recommendation:**

Weak Reject: I recommend rejecting the paper, but will not argue for my recommendation if the majority of other reviewers have a different opinion.

**Summary:**

This paper presented a new method to extend the Dense Object Nets (DON) to multi-object scenes. It presents two ways to learn class-aware dense object descriptors to be robust to multi-object scenes.

**Issues:**

In addition to the issues mentioned above, I will also be interested to see if the following concerns can be addressed.

This paper shows that the trained feature descriptor can separate different object classes. It would be interesting to see the feature descriptor visualizations and also compare them with DON's feature descriptor visualization. It can bring more insights, as DON did in their paper.

In the TSNE visualization for the soft labelling, it seems that the sprite can and pepsi can feature descriptor embeddings have mixed with the book embeddings. It means the book and can categories have mixed. I am not very convinced these two instances share similarities with the books unless some other (e.g. visual or qualitative) evidence can be shown. I am also wondering if this issue can be mitigated by increasing the feature descriptor dimensions. Actually, the authors did not mention the feature dimension number they use for the model. It is an important hyperparameter and can influence performance a lot.

The only baseline method this paper compared to is DON. However, there have been many papers coming out in the field of learning feature descriptors, such as Self-supervised visual descriptor learning for dense correspondence, sosnet, d2net, or HyNet. Most of those approaches also learn the feature descriptor based on triplet loss or contrastive loss. It would be interesting to see some comparisons to these generic feature descriptor methods. To obtain the per-pixel correspondence from these descriptor approaches, one can simply run an argmin of l2 feature descriptor distance to the target image, as done in the DON and its following works.

As the authors have talked about in the limitations section, the usage of K-means has constrained the application of this approach. Knowing the number of clusters in advance is a bit contradictory to the target usage case of multi-object scenes.

In terms of the writing part:
* equation 5: What does c.M mean?
* L120: object view variance: has it already been addressed by training on the images captured from different viewpoints?

**Reviewer Expertise:**

Very good: Comprehensive knowledge of the area

**Strengths And Weaknesses:**

The main and the only baseline, which this paper has compared to, is the Dense Object Nets (DON). Compared to DON, this paper shows the robust feature matching in the multi-object case. This innovation in this paper is a bit incremental but it shows a step further to the real-world applications.

My main concern is that the testing case in this paper is a bit constrained.
Although it targets multi-object scenes, the test scenes actually only contain one instance per class. It would be interesting to test how the method will work in an environment where multiple shoes or hats exist. This would be a more realistic scenario.
Secondly, it seems that the reference image in the test has been seen in the training (L209, L228). This makes the test cases weak. It will be more convincing if the method can use an unseen object (in the same category) for the reference image.
Thirdly, the method only trained for two-class. It will be important to see if such binary classification can be generalized to more classes.


**Summary Of Recommendation:**

The innovation in this paper is limited. It also requires some further improvements in the writing and experiments, even if it shows some interesting experimental results. My recommendation opinion is based on these considerations.

---

> ### Author Response · Authors · 2021-08-23
> **Comment Answer to Reviewer ZCDs - Part 1**
>
> **Summary of major changes:**
> - Changed title to "Fully Self-Supervised Class Awareness in Dense Object Descriptors''
> - Major rewording of Section 4.2.1 Similarity Graphs to be more clear and informative.
> - Additional figure showing the structure of the data is added in Section 4.1. Problem Setup.
> - Experimental section now explicitly separates the setup and results of each experiment. Data is summarized in a table and experiments state which data they use.
> - The system figure (Fig. 3 in revision) has been changed to include all outputs and be more clear, rather than abstract.
> - Some references have been added to related work per suggestion of the reviewers - (Pirk et al., Schmidt e al., Grasp2Vec)
> - Experiments with a lower and upper baseline were ran for the quantitative evaluation. These have been shown in large scale in the supplementary materials.
> - To fit these changes in the paper's 8-page limit, some sections were re-ordered, trimmed and changed. Further figures answering reviewers concerns have been added to the supplementary files.
>
> Dear Reviewer, we would like to thank you for the review. Below, we have addressed your comments in order to answer your concerns and improve the quality of our work.
>
> >My main concern is that the testing case in this paper is a bit constrained.
> Although it targets multi-object scenes, the test scenes actually only contain one instance per class. It would be interesting to test how the method will work in an environment where multiple shoes or hats exist. This would be a more realistic scenario.
>
> The Reviewer is correct in pointing out that a more realistic scenario would contain multiple objects of the same category. With the way that the system / evaluation is currently set up, a best match is found by L2 distance as in Florence et al. (CoRL 2018): this is not ideal and wouldn't work great if there are multiple objects to match with. The model finds a correspondence on one of the objects, until it is removed; then proceeds to find it on a different one. If we put a threshold of sorts and find ALL matches that satisfy it, perhaps there would be natural clusters of points and thus multiple correspondences can be found. However, this was not implemented or tested for. Some of the multi-object test images do contain multiples of objects that were trained (e.g. 2 shoes), but only when being tested on other objects (e.g. hat).
>
> >Secondly, it seems that the reference image in the test has been seen in the training (L209, L228). This makes the test cases weak. It will be more convincing if the method can use an unseen object (in the same category) for the reference image.
>
> We imagined a scenario in which the robot recalls objects that it has already learned, and uses that memory to find corresponding points on novel objects at inference time. Thus, the reference images for those tests were taken from the training set, while query images are novel (with potentially unseen objects).
>
> > Thirdly, the method only trained for two-class. It will be important to see if such binary classification can be generalized to more classes.
>
> This has been answered in Reviewer's 2SMq comments (see the corresponding answer). In brief, please see the revised Fig. 7 (in the new draft) where we show how does the method performance changes when working with more classes (5). Notice that we are limited by the amount of classes present in the real dataset that were used by Florence et al. (CoRL 2018) as we had to use object categories that have multiple different instances.
>
> > The innovation in this paper is limited. It also requires some further improvements in the writing and experiments, even if it shows some interesting experimental results.
>
> We would like to stress that the impact of the paper is on removing the limitation of the original DONs (Florence et al. - CoRL 2018) to work with only single objects. Our proposed work led to a fully self-supervised learning method that can extend in using those descriptors in the wild.  Our work's novelty lies in extending on the potential applicability of these methods when being used to multiple object categories; while the original original CoRL 2018 paper alone, only works with a single category of objects.
>
> >This paper shows that the trained feature descriptor can separate different object classes. It would be interesting to see the feature descriptor visualizations and also compare them with DON's feature descriptor visualization. It can bring more insights, as DON did in their paper.
>
> Given the space limitation, some examples of feature descriptor images have been added to the supplementary materials (Fig. 5-Supplementary).

---

> > ### Author Response · Authors · 2021-08-23
> > **Comment Answer to Reviewer ZCDs - Part 2**
> >
> >
> > >In the TSNE visualization for the soft labelling, it seems that the sprite can and pepsi can feature descriptor embeddings have mixed with the book embeddings. It means the book and can categories have mixed. I am not very convinced these two instances share similarities with the books unless some other (e.g. visual or qualitative) evidence can be shown.
> >
> > Supplementary materials now contain figures showing the samples of pepsi/sprite compared with books (Fig. 4-Supplementary). There are some books that have similar colours/textures, so it is understandable why a cluster-based method like DON+Soft's similarity graph may be connecting them in its similarity graph. The Reviewer is correct in pointing this out, and it is interesting to investigate whether additional information such as the depth or geometry can also be used in the creation/population of the similarity graph - to avoid such 'wrong' connections.
> >
> > >I am also wondering if this issue can be mitigated by increasing the feature descriptor dimensions. Actually, the authors did not mention the feature dimension number they use for the model. It is an important hyperparameter and can influence performance a lot.
> >
> > We would like to thank the Reviewer for pointing this out. The number of dimensions (D=5) have now been added to the hyperparameter section.
> >
> > >The only baseline method this paper compared to is DON. However, there have been many papers coming out in the field of learning feature descriptors, such as Self-supervised visual descriptor learning for dense correspondence, sosnet, d2net, or HyNet. Most of those approaches also learn the feature descriptor based on triplet loss or contrastive loss. It would be interesting to see some comparisons to these generic feature descriptor methods. To obtain the per-pixel correspondence from these descriptor approaches, one can simply run an argmin of l2 feature descriptor distance to the target image, as done in the DON and its following works.
> >
> > We would like to stress that the approaches mentioned by the Reviewer do not use semantic correspondences, but rather direct correspondences. However, for a similar baseline we included DAISY descriptors and DON+Ground truth labels (as in Florence et al.'s specific method with cross-object loss - CoRL 2018), as well as DON+Soft+Ground Truth in the supplementary materials (Fig. 1- Supplementary).
> >
> > > As the authors have talked about in the limitations section, the usage of K-means has constrained the application of this approach. Knowing the number of clusters in advance is a bit contradictory to the target usage case of multi-object scenes.
> >
> > As noted in Sec. 6-Limitations, this constraint does not affect DON+Soft, which performs better than DON+Hard: `"DON+Soft would be unaffected by this since it purely relies on similarity measures."` Similarity measures do make use of K-Means, but their cluster sizes are fixed and do not depend on the number of classes.
> >
> > > What does c.M mean?
> >
> > c is the confidence parameter defined in Similarity Graph section. In the revised paper this is written as `"We propose a dissimilarity confidence $c$ in the pairing equal to the min-max normalized $W^-$."` M is the margin parameter used by metric learning triplet losses. It is defined in the hyperparameter section as 0.5 (i.e., we noted M as the margin parameter). Since the confidence is bigger when W- is bigger, it reflects how confident we are that the pairing is dissimilar. The term c.M 'moves' the lower limit of the ReLU depending on the confidence in the pairing - thus pairs that we are confident in have bigger loss and vice versa.  The revised phrasing of this should make things clearer.
> >
> > > L120: object view variance: has it already been addressed by training on the images captured from different viewpoints?
> >
> > As the Reviewer points out, we assume we know the robot pose of each frame view (via the fixed-based manipulator's kinematics), thus we can address this by explicitly grouping by viewpoints and it is an interesting direction to explore.  However, following Tan et al. (ICRA 2021), we used a clustering to address the viewpoint problem. While it is potentially more computationally expensive, this approach may also offer other advantages such as clustering together views that are semantically similar due to object symmetries and addressing viewpoint variances too.

---

> > > ### Comment · Reviewer_ZCDs · 2021-09-02
> > > **Thanks for solving my concerns, however, the applicability of this approach is still in doubt**
> > >
> > > I would like to thank the authors for making efforts to generate data and improving the writing to solve my concerns. Most of my concerns are addressed in their responses. However, I am still unsure about the applicability of this approach in multi-object test cases which prohibits me from change opinions.
> > >
> > > The experiments video still only contains one instance per class. The performance drop of multi-object matches on 5-classes in Figure 7 as well as the mixed T-SNE of DON+Soft concern me about its capability in working with multi-object scenarios.
> > >
> > > Besides, as the author said in the response, the current setting probably cannot work if there are multiple instances that belong to the same category in the scene. So it is a bit different from what is claimed in the abstract and introduction.
> > >
> > > Despite these, I agree this work is an improvement from DON and can make some contributions to the CoRL. So as the Recommendation said, "I will not argue for my recommendation if the majority of other reviewers have a different opinion.".

---

### Official Review · Reviewer_2SMq · 2021-07-26

**Originality:** Good
**Technical Quality:** Good
**Clarity Of Presentation:** Good
**Impact:** 2

**Recommendation:**

Weak Accept: I recommend accepting the paper, but will not argue for my recommendation if the majority of other reviewers have a different opinion.

**Summary:**

The paper extends the Dense Object Network (DON) method by incorporating object category awareness. The idea of DON is to learn pixel-wise object descriptors in order to detect pixel-to-pixel correspondences between two objects of the same category appearing in different images. While DON focuses on finding correspondences between objects in single-object scenes, this paper extends DON to cope with multi-object scenes. The idea is to generate object category labels for each object in an unsupervised fashion (through clustering in a feature space), and use these labels to inform the object descriptor learning. The experiments show the promising results for disentangling two object categories in a multi-object scene and also improves the single-object case.


**Issues:**

The paper would make a stronger point if more object classes in particular with similar shapes would be considered during evaluation and training.



[The following issues concern an earlier version and have been addressed in the paper revision]

Technical approach:
- It is a bit hard to follow the distinction between instance classes, categories, sequences, random walks on the graph etc. A figure giving an overview of all these entities and how they fit together would be extremely helpful.
- The paper combines DONs with similarity graphs, and as both are critical components, similarity graphs should be introduced at the same level as DONs. Furthermore, it would be better to first explain the general idea of these similarity graphs (L137-L140) before delving into the details. Finally, a figure explaining how random walks on these graphs look like, how the matrices W+ and W- come into play etc. would be helpful.
- The unsupervised generation of labels seems quite complicated, in particular since only two object classes are present. The paper doesn't fully convince me that such complex label generation machinery is necessary.
- I can think of one simple lower baseline, for example obtaining labels by classifying the images with ImageNet und using ImageNet labels instead of the unsupervised approach, but I believe there are even easier unsupervised ones.
- Similarly, it would be interesting to consider an upper baseline, by providing the ground truth label of the item class explicitly to the network.

Experimental evaluation:
- It is stated on the side that only two types of objects categories per experiment are considered. The description of the dataset, incl. how many instances are used, etc. should be best summarized in an experimental setup section and a table.
- Entanglement of hypotheses and result discussion: How the experimental conditions differ between experiments 5.1, 5.2, 5.3? Are the same objects, trained networks etc used? This is not fully clear to me
- L184: Why are K=2 and K=8 chosen? Why not K=3, 4, 20 ...?
- L179 states a detail about 500 training steps - what is that mentioned here, why does it matter?
- L172-173 states that the objective of the experiment is to “answer effective our method is at separating object descriptors” - but isn’t the scope of the paper broader, namely to present a method for class-aware dense object matching?
- L178: “consider three input types” - input to what? Do you mean this is the feature space used for k-means?

Technical questions:
- How is the depth/3D image used? The paper doesn’t mention any details about why a 3D sensor is required at all for descriptor learning.
- L140: “Complicated” per se does not mean that something is bad. Maybe it performs better than what is proposed here? Why was the original approach discarded?
- L165: How are these sub-loss functions L_pos,non-match, etc. defined?
- L158: “a negative sequence pairing is randomly chosen” - how exactly? By starting from the randomly sampled sequence and then…?

Minor issues:
- In the introduction, I would suggest to explain on simple terms and concisely the high-level intuition of the method. E.g. the term “class awareness” is used in the title yet “unsupervised classification” is introduced in L21 without explaining how this unsupervised classification is going to work.
- L90 “whereas” doesn’t fit here
- L104 mentions “background augmentation” is mentioned but not explained before
- Figure 3: I would suggest to add the inputs and outputs as well, I was confused by what the output of “Projection” is


**Reviewer Expertise:**

Good: General knowledge of the area

**Strengths And Weaknesses:**

## Strengths

The paper addresses the important problem of finding dense correspondences between objects and object parts without CAD models nor explicit object category labels. The paper shows promising results for disentangling two object categories (cans vs. books and hats vs. shoes) in a multi-object scene. The technical approach is sound and the presentation good. The example visualizations given in the paper are illustrative of the task.

## Weaknesses

The main weakness of the paper are that the method is only evaluated on a low number of object classes. It is difficult to judge how generalizable the method is given that descriptors are only learned on two / five classes of objects respectively. I would expect the approach to be evaluated on more objects, including such object with similar shapes.

**Summary Of Recommendation:**

The paper tackles an important problem, provides a good presentation and a technically sound approach. One weakness is that few object classes are used for evaluation which renders the experiments somewhat preliminary and limits the contribution of the paper.

---

> ### Author Response · Authors · 2021-08-23
> **Comment Answer to Reviewer 2SMq - Part 1**
>
> **Summary of major changes:**
> - Changed title to ''Fully Self-Supervised Class Awareness in Dense Object Descriptors''
> - Major rewording of Section 4.2.1 Similarity Graphs to be more clear and informative.
> - Additional figure showing the structure of the data is added in Section 4.1. Problem Setup.
> - Experimental section now explicitly separates the setup and results of each experiment. Data is summarized in a table and experiments state which data they use.
> - The system figure (Fig. 3 in revision) has been changed to include all outputs and be more clear, rather than abstract.
> - Some references have been added to related work per suggestion of the reviewers - (Pirk et al., Schmidt e al., Grasp2Vec)
> - Experiments with a lower and upper baseline were ran for the quantitative evaluation. These have been shown in large scale in the supplementary materials.
> - To fit these changes in the paper's 8-page limit, some sections were re-ordered and trimmed. Further figures answering reviewers concerns have been added to the supplementary files.
>
> Dear Reviewer, we would like to thank you for the review. Below, we have addressed your comments to improve the presentation of the technical approach and experimental setup of our work.
>
> > Technical approach: It is a bit hard to follow the distinction between instance classes, categories, sequences, walks on the graph etc.
>
> To make clearer the distinction between different entities (i.e., instance classes, categories, sequences, and graph walks) in the paper:
>
> 1) We explained the difference between **instance classes** and **categories** in the paper: `''e.g. different hats would belong to different instance classes but the same category.''`
>
> 2) We have also added a figure in supplementary materials (Fig. 3-Supplementary) showing abstractly how **instances** are different from **categories**.
>
> 3) As suggested by the Reviewer, we added a figure explaining the data structure (Fig. 2 in revision). As it can be seen in the newly added figure, a **sequence** is comprised of all **datapoints** for a single object over time. An **instance class**, e.g. a shoe, can be present in multiple **sequences**. A **sequence** is simply the whole continuous sequence of **datapoints** when an object is observed.
>
> 4) To explain better the **Random Walks on Graphs** we improved the text in Sec. 4.3.2. (which is Sec. 4.2.1. in the revision), particularly in the first and last paragraph: `''RWS is the sampling of node pairs by ''walking'' on edges with transitional probability proportional to the edge weights.''`  and also `''Given a starting node in the graph, a transition probability distribution $p_t$ is formed proportional to the connected edge weights. By sampling a transition from $p_t$, a 'random walk' is done to a node that is more likely similar. Similarly, sampling from $(1-p_t)$ is used to reach more probably dissimilar nodes.''`
>
> >Why are similarity graphs needed in the first place? Isn’t there a simpler way to obtain object-identity labels and use those to train the descriptors? The paper combines DONs with similarity graphs. It provides a good explanation of DONs, but the explanation of similarity graphs, why they are chosen and how they work is somewhat obscure.
>
> We have reworded the section to address the issues mentioned by the Reviewer and improve the clarity of the section. A paragraph is added in Sec. 4.3.2 (which is Sec. 4.2.1 in revision) to address both the motivation of similarity graphs, as well as how they work.
>
> In brief, to answer **why** Similarity Graphs: since we have no access to ground truth labels (i.e., since we assume that ground truth labels are unavailable for self-supervision), we needed to somehow produce our own labels - after reviewing methods that disentangle object representations and are unsupervised or self-supervised, we settled on basing our method on Tan et al.'s Similarity Graphs (ICRA 2021) for the sole reason that it showed recent State-of-the-Art results. While the label generation method's mathematical formulation may seem convoluted, we believe its implementation is relatively simpler than other reviewed methods (e.g. using geometric information).
>
> Briefly, to answer **how** Similarity Graphs work: a Graph is constructed with nodes that are "sequences of objects masks in consecutive frames", while the weights of each edge are "the degrees of similarities between the sequences". To compute the degree of similarity one needs to search the "best  alignment  of  the  different  viewpoints  of  the  objects  in  two sequences", i.e., the transition probabilities.  Then, a random  graph walk creates examples of similar or dissimilar objects from different image sequences (see Fig. 1 in Tan et al. - ICRA 2021).

---

> > ### Author Response · Authors · 2021-08-23
> > **Comment Answer to Reviewer 2SMq - Part 2**
> >
> >
> > >Experimental evaluation: The experimental section entangles the hypotheses to be studied with technical details of how the experiments were conducted. The experimental setup should be clearly separated from the results and discussion.
> >
> > Following the Reviewer's suggestion, the experiments (Sec. 5) have been more clearly separated into **Setup** and **Results** in the new revision.
> >
> > > It is difficult to judge how generalizable the method is given that descriptors are only learned on two classes of objects. I would expect the approach to be evaluated on more objects, including such object with similar shapes.
> > > Also, considering only two object classes for evaluation renders the experiments somewhat preliminary and limits the contribution of the paper.
> > > The unsupervised generation of labels seems quite complicated, in particular since only two object classes are present. The paper doesn't fully convince me that such complex label generation machinery is necessary.
> >
> > To understand how generalizable the method is, we have conducted an experiment evaluating how much information is retained when increasing the number of class categories on which the model is trained. **In particular, we increased the category classes from 2 to 5, by also increasing the different instance classes. The resulting drop in performance in our multi-object quantitative evaluation is almost unnoticeable. See Fig. 6 (which is Fig. 7 in revision).**
> >
> > Moreover, it is worth noting that our work extends the one of Florence et al. (CoRL 2018) and uses their real data for fair comparison. For our method, we are limited to using the category classes they provided that have comparable numbers of different instance classes and were visually similar enough (only hats and shoes satisfied both of these conditions). In the future, we plan in future to generate our own real-world data and extend our method to more class categories.
> >
> > > The unsupervised generation of labels using similarly graphs seems quite complicated, and the paper is missing both a simpler lower baseline as well as an upper baseline (ground truth object labels).
> >
> > Following the Reviewer's suggestion, in the supplementary materials (shown large in Fig. 1-Supplementary) we added a graph with a quantitative experiment, extending the revised Fig. 7 of the paper, which compares also with DON+GT trained as in Florence et al. (CoRL 2018), DON+Soft+GT which is our DON+Soft model trained with ground truth pairing confidence, and DAISY descriptors for lower baseline.
> >
> > **The outcomes of this new added comparison are as follows:**
> > 1) As expected from any descriptor that does not have semantic awareness, DAISY features do not perform well.
> > 2) DON+GT, which is DON trained with Ground truth labels, works with 'hard' discrete labels and performs similarly to DON+Hard, likely because these labels are only used to 'repel' descriptors without any descriptors to 'attract' them to, i.e. **without contrastive learning, these discrete labels are not useful**. In comparison, DON+Soft uses contrastive learning directly on the descriptors - it attracts positive same-sequence matches while simultaneously repelling the likely negative match.
> > 3) The other upper bound we added - DON+Soft+GT where we set the confidence to 1 or 0 depending on Ground truth pairing performs similarly to DON+Soft in multi-object scenarios.
> > 4) For single-object matches, both ground truth upper bounds (DON+GT and DON+Soft+GT) significantly outperform the other methods.
> >
> > > It is a bit hard to follow the distinction between instance classes, categories, sequences, random walks on the graph etc. A figure giving an overview of all these entities and how they fit together would be extremely helpful.
> >
> > This has been handled in the reviewer's "Technical Approach" comment above.
> >
> > >The paper combines DONs with similarity graphs, and as both are critical components, similarity graphs should be introduced at the same level as DONs. Furthermore, it would be better to first explain the general idea of these similarity graphs (L137-L140) before delving into the details.
> >
> > Following the reviewer's suggestion to clear this part up, a paragraph (revised Sec. 4.2.1) explaining and motivating the use of Similarity Graphs before jumping onto the technical details. At the end of the section, there's an additional reworded paragraph explaining random walks on graphs.

---

> > > ### Author Response · Authors · 2021-08-23
> > > **Comment Answer to Reviewer 2SMq - Part 3**
> > >
> > >
> > > >Finally, a figure explaining how random walks on these graphs look like, how the matrices W+ and W- come into play etc. would be helpful.
> > >
> > > A figure (Fig. 2 - Supplementary) has been added in the supplementary materials. The W+ and W- are measures of how similar and dissimilar each pair of nodes (each node representing a sequence) are - while W+ measures similarity based on how often the sequences share similar global clusters (i.e. their features are more similar to each other than to other nodes), W- measures dissimilarity as the distance between the sequences' local clusters. The intuition is that if two sequences represent a similar object, their local cluster centroids will be similar as well.  All of these measures are using the features outputted by a pre-trained ResNet embedder. The text in the paper draft has been rephrased to make it clearer in the revision.
> > >
> > > >I can think of one simple lower baseline, for example obtaining labels by classifying the images with ImageNet und using ImageNet labels instead of the unsupervised approach, but I believe there are even easier unsupervised ones.
> > > >Similarly, it would be interesting to consider an upper baseline, by providing the ground truth label of the item class explicitly to the network.
> > >
> > > We would like to thank the Reviewer for the suggestions. Indeed, we did train with ground truth labels, as it is also done in the Florence et al. (CoRL 2018) paper.  However, we found the performance to be similar to that of DON+Hard. We believe the softness of the DON+Soft labels is one of the major contributors to the performance increase, i.e., some objects such as boots, are both similar and dissimilar to other shoes in their class, the other factor being the contrastive loss applied on the descriptors (rather than projection network of DON+Hard). Due to not performing well as an upper baseline, we excluded it to not make those graphs too cluttered and hard to read. For a lower baseline we have added a dense descriptor that does not consider semantic information - DAISY, found in supplementary materials (Fig. 1-Supplementary).
> > >
> > >
> > > >It is stated on the side that only two types of objects categories per experiment are considered. The description of the dataset, incl. how many instances are used, etc. should be best summarized in an experimental setup section and a table.
> > > > Entanglement of hypotheses and result discussion: How the experimental conditions differ between experiments 5.1, 5.2, 5.3? Are the same objects, trained networks etc used? This is not fully clear to me
> > >
> > > We have now explicitly separated the **Setup** and **Results** for each experiment. The dataset is summarized in a table (Table 1) including nicknames to each part of the dataset which are then specified in each experiment. In Sec. 5 we have identified more clearly the difference between experiments and conditions, as well as the type/number of objects and networks that are used. We sum-up below the objects that are used:
> > > - Similarity Graph experiment: models are trained both with simulated (Cans and Books) and real data (Hats and Shoes)
> > > - TSNE experiment: models are trained  with simulated data (Cans and Books)
> > > - Accuracy experiments: models are trained with real data (Hats and Shoes), except for the 5-class experiment that models are trained with extra real data (Hats and Shoes, and Robot toys)
> > > - Real experiments - models are trained with real data (Hats and Shoes)
> > >
> > > >L184: Why are K=2 and K=8 chosen? Why not K=3, 4, 20 ...?
> > >
> > > We chose K=2 as that is the ground truth number of object classes, and K=8 as that is the ground truth number of instance classes. We have added more information in the paper draft for clarity. We did not use any other value for K because it does not correspond to any corresponding object or instance class number, thus we cannot assess the ''correctness'' of the clustering.
> > >
> > > > L179 states a detail about 500 training steps - what is that mentioned here, why does it matter?
> > >
> > > The 500 training steps mean that for Projection Network is trained for 500 steps before proceeding to the phase-2 training. It is added there for reproduction of our results. We have reworded this to be more clear in the draft.
> > >
> > > > L172-173 states that the objective of the experiment is to “answer effective our method is at separating object descriptors” - but isn’t the scope of the paper broader, namely to present a method for class-aware dense object matching?
> > >
> > >  Our goal is to produce fully self-supervised descriptors that can correctly find matches between semantically similar objects (i.e. class-aware) in multi-object scenes. Since the multi-object scenes could contain multiple objects on which the model is trained on, it should have the ability to disentangle (separate) the descriptors. For this reason, we produced the qualitative TSNE projection, but more importantly the qualitative results presented after the TSNE in the draft.

---

> > > > ### Author Response · Authors · 2021-08-23
> > > > **Comment Answer to Reviewer 2SMq - Part 4**
> > > >
> > > >
> > > > > L178: “consider three input types” - input to what? Do you mean this is the feature space used for k-means?
> > > >
> > > >  Yes, we mean the input features used by the Similarity Graph. We have reworded the Similarity Graph section to explicitly talk about the chosen input features and reference to this experiment to improve clarity.
> > > >
> > > > > How is the depth/3D image used? The paper doesn’t mention any details about why a 3D sensor is required at all for descriptor learning.
> > > >
> > > >  As our method is using the DON as a basis, we explained in L66-L68 of the originally submitted paper, that `''It uses the strong prior of depth information taken from an RGB-D camera, and knowledge of a robot manipulator’s pose to project 2D image pixels onto a reconstructed 3D model, thus learning correspondences between different views of the same 3D points.''`. We added this also in the methodology section for more clarity.
> > > >
> > > > > L140: “Complicated” per se does not mean that something is bad. Maybe it performs better than what is proposed here? Why was the original approach discarded?
> > > >
> > > >  We discarded the original confidence simply because it didn't work - it can work with correct tuning of the lambda parameter, however, we found that if we use the Tan et al. (ICRA 2021) confidence, this parameter is hard to tune if we don't have uniformly frequent data (e.g. equal amounts of sequences containing boots and sneakers). Our proposed method, reduces this requirement significantly by using normalizations and removing it altogether from the confidence equation. We have reworded this part of the text in the revised draft.
> > > >
> > > > > L165: How are these sub-loss functions L_pos,non-match, etc. defined?
> > > >
> > > > These losses refer to the losses defined in equations (1) and (2). These references are now written explicitly. We would like to thank the Reviewer for pointing this out.
> > > >
> > > > > L158: “a negative sequence pairing is randomly chosen” - how exactly? By starting from the randomly sampled sequence and then…?
> > > >
> > > >  We have reworded the similarity graph section (revised Sec. 4.2.1), including a paragraph on random walking. `"Given a starting node in the graph, a transition probability distribution $p_t$ can be formed proportional to the connected edge weights. By sampling a transition from $p_t$, a 'random walk' is done to a node that is more probably similar. Similarly, sampling from $(1-p_t)$ is used to reach more probably dissimilar nodes."` This is the way walks are done in any other mentions.
> > > >
> > > > > In the introduction, I would suggest to explain on simple terms and concisely the high-level intuition of the method. E.g. the term “class awareness” is used in the title yet “unsupervised classification” is introduced in L21 without explaining how this unsupervised classification is going to work.
> > > >
> > > > We removed the mention of ''unsupervised classification'' there to remove confusion. It is followed by an explanation of the two variants approaches: discrete labelling and soft confidence scores. Change in the title (as suggested by Reviewer RTPV) to `Fully Self-Supervised Class Awareness in Dense Object Descriptors` further emphasizes on the fact that the whole process requires no external supervision.
> > > >
> > > >  >  L90 “whereas” doesn’t fit here
> > > >
> > > >  This has been rephrased.
> > > >
> > > > > L104 mentions “background augmentation” is mentioned but not explained before
> > > >
> > > > A line has been added to Sec. 3 Dense Object Nets: `''Augmentations consist of random background, color jitter, crops, and scale''`. The original DON method uses the object masks to remove the background and replace it with something random like a colour gradient (domain randomization). It also does jittering, cropping, rescaling, flipping - transformations after which we can still track where the 3D points are in relation to the 2D image.
> > > >
> > > > > Figure 3: I would suggest to add the inputs and outputs as well, I was confused by what the output of “Projection” is
> > > >
> > > > We would like to thank the reviewer; we have now added the inputs and outputs too.

---

> > > > > ### Comment · Reviewer_2SMq · 2021-09-03
> > > > > **Thank you**
> > > > >
> > > > > I would like to thank the authors for thoroughly revising the paper and addressing my concerns. My only concern left is that the results are still a bit preliminary given the overall low amount of object classes, but I think the paper provides sufficient evidence that the approach is promising. The revision also significantly improved the presentation of the paper. I'm therefore changing my vote to weak accept.

---

### Meta-Review · Area_Chair_LDZa · 2021-08-16

**Recommendation:** Accept (Poster)
**Confidence:** 4

**Metareview:**

This paper extends DON by making the object labels self-supervised and found via cluster assignment in pretrained visual feature space. This removes the prior limitation of DONs where the object-in-view needed to be labeled. The authors also show that augmenting the training set with cluttered images also improves generalization to multi-object scenes.

This paper currently has 2 weak rejects and 1 strong accept, with reviewer RTPV willing to go to bat for this paper.

*Strengths*: Reviewers appreciate the focus on scalable data for representation learning. Figure 6 represents an improvement over DON for multi-object matches, real robot results

*Weaknesses*: Reviewers 2SMq, RTPV say that the motivations and technical implementation details of Similarity Graphs (Sec 4.3.2) are unclear. 2SMq and ZCDs also recommend that the overall method be improved for clarity. Both 2SMq and ZCDs take issue with the test scenes only involve two objects classes and only contain one instance per class. I think this is a valid criticism, because it is quite possible that a cluster-based assignment for auto-labeling data may scale poorly as the number of classes is increased.

I have some questions about Figure 6 that I would like the authors to respond to in their rebuttal, and potentially RTPV as well.

It seems to me from reading the results in Figure 6 that the improved generalization to multi-object scenes is not a definitive property of the method, but rather due to the use of incorporating multi-object scenes into the training data (L208), and the use of soft labels. The benefit of training on multi-object scenes is obvious, so the remaining question is whether the proposed method contributes anything new to better descriptors in multi-object scenes beyond the data. Figure 6 seems to suggest this, but I noticed that the DON baseline performs about as well as the hard variant of the method on multi-object matches (hard to really understand the relative difference between blue/orange in Fig 6). This leads me to suspect that the improved performance of the soft labels is not due to the clustered label assignments, but rather the fact that the labels are just soft. In other words, a null hypothesis is that "one does not require a self-supervised method to recover soft labels; one could imagine a simple label smoothing trick applied to the hard labels would work as well as the DON + Soft Labels".

I would appreciate if the authors could check this hypothesis. Another experiment  to check the null hypothesis would be to train on self-supervised labels from single-object scenes only, and then test on multi-object scenes. If my suspicions are correct, both DON and DON + self-supervised labels will do equally poorly.

If the null hypothesis that "soft labels improve multi-object generalization for DON" is a property of the label softness and not the self-supervised clustering method itself, I think that does diminish the novelty of the paper, especially considering that the method is simply using clustering to do label propagation. Many prior works such as Caron et al. '18 have utilized this trick to provide labels for some other downstream learning task. Connecting unsupervised label propagation to a relatively new descriptor learning method is not guaranteed to "just work", so I appreciate the research being done here, but then I think some more compelling evidence around the promised scalability of the method needs to be shown. See RTPV's point about "across or between different scenes is critical" or handling more than 2 object classes as the other reviewers have pointed out. One might argue that it would not be hard to extend the existing DON collection protocol (e.g. a camera orbiting a fixed object for which you know the class) by tossing in some distractor objects to make the data multi-view without additional labeling cost.

Some relevant work the authors may want to look at:
- There have been some prior works like Grasp2Vec (Jang and Devin et al. 2018) that show dense, self-supervised object features learned in multi-object scenes (though not via pixel-to-pixel correspondences) and Object Contrastive Nets (Pirk et al. 2019) which also propose self-supervised object correspondences in multi-object scenes. Pirk et al. uses a single scene to learn object representations, but conceptually may not be limited by that.


---- update 20210904

I thank the authors for their detailed rebuttal and for the reviewers in engaging in discussion. One of the reviewers have updated their rating to weak accept, and given that the authors have addressed all my concerns as well, I will recommend an accept.

---

> ### Author Response · Authors · 2021-08-23
> **Comment Answer to Meta-Reviewer**
>
> ﻿**Summary of major changes:**
> - Changed title to "Fully Self-Supervised Class Awareness in Dense Object Descriptors''
> - Major rewording of Section 4.2.1 Similarity Graphs to be more clear and informative.
> - Additional figure showing the structure of the data is added in Section 4.1. Problem Setup.
> - Experimental section now explicitly separates the setup and results of each experiment. Data is summarized in a table and experiments state which data they use.
> - The system figure (Fig. 3 in revision) has been changed to include all outputs and be more clear, rather than abstract.
> - Some references have been added to related work per suggestion of the reviewers - (Pirk et al., Schmidt e al., Grasp2Vec)
> - Experiments with a lower and upper baseline were ran for the quantitative evaluation. These have been shown in large scale in the supplementary materials.
> - To fit these changes in the paper's 8-page limit, some sections were re-ordered and trimmed. Further figures answering reviewers concerns have been added to the supplementary files.
>
> Dear Meta-Reviewer, we would like to thank you for the review and the suggestions. Below, we have addressed your comments.
>
> >The authors also show that augmenting the training set with cluttered images also improves generalization to multi-object scenes.
>
> We believe that there was a misunderstanding. We did not augment the training set with cluttered images; we only used single object images. In L91-93 of the original paper we state `"In this paper, each training sequence strictly contains a single instance of a single object, while during inference we can work with both single and multi-object sequences.  We discuss how the  method  could  be  adapted  to  train  from  multi-object  sequences  in  Sec.  6."` Text in the revised paper draft has been updated to emphasize this and make it clearer.
>
> >Reviewers 2SMq, RTPV say that the motivations and technical implementation details of Similarity Graphs (Sec 4.3.2) are unclear. 2SMq and ZCDs also recommend that the overall method be improved for clarity.
>
> The implementation details for similarity graphs are reworded for clarity and understanding. An additional paragraph motivating the use of similarity graphs and explaining them before technical details is also added (Sec 4.3.1 of revised version).
>
> > the test scenes only involve two objects classes and only contain one instance per class
>
> Please see our answers to Reviewer 2SMq and ZCDs. We ran most experiments with two object classes containing a total of 8 instances. In the revision, we emphasized the number of instance classes (8 for simulated and 8 for real data) as it was not easy to read in the original submitted paper. We have updated the paper to add this information in the Training Data section (Sec. 5, Table 1).
>
> As for training on more than two classes: please note the brief experiment that appear in Sec. 5 ("Descriptor Match Accuracy Experiments" and Fig. 7 in revision), has shown that the increase of classes produced comparable results. We agree with the Meta-Reviewer and we do believe that the number of classes that can be learned may be indeed limited by any particular choice of parameters. This is a problem that could potentially be solved by increasing the number of descriptor dimensions and the cluster parameters of the similarity graph. As we are using the dataset of real images provided by the Florence et al. paper (CoRL 2018) we were limited in the amount of data that we could test on and compare with for this experiment.
>
> > It seems to me from reading the results in Figure 6 that the improved generalization to multi-object scenes is not a definitive property of the method, but rather due to the use of incorporating multi-object scenes into the training data (L208) ...
>
> Please note that we did not use multi-object images as part of the training. The ability of the model to work with multiple classes in the scene is purely in evaluation only, during inference.
>
> >In other words, a null hypothesis is that "one does not require a self-supervised method to recover soft labels; one could imagine a simple label smoothing trick applied to the hard labels would work as well as the DON + Soft Labels".
>
> As explained in the revised Fig. 4 and Sec. 4.2.2, that the DON+Hard labels are created via proxy of the projection network, which is learned similarly to the method suggested in DON+Soft. The hard labelling system is actually more complex than the soft labeling one as it itself uses similarly created soft labels to produce the hard labels. Thus applying a soft labeling trick on DON+Hard labels that were generated via soft labeling seems to be unnecessary. Moreover, our soft confidences are based on feature similarity unlike the smoothing trick as proposed by Szegedy et al. (CVPR, 2015)

---

> > ### Author Response · Authors · 2021-08-23
> > **Comment Answer to Meta-Reviewer - Part 2**
> >
> >
> > >Another experiment to check the null hypothesis would be to train on self-supervised labels from single-object scenes only, and then test on multi-object scenes. If my suspicions are correct, both DON and DON + self-supervised labels will do equally poorly.
> >
> > As written in previous comments, the models  were indeed trained only on single-object images and tested on multi-object images already.
> >
> > > If the null hypothesis that "soft labels improve multi-object generalization for DON" is a property of the label softness and not the self-supervised clustering method itself, I think that does diminish the novelty of the paper, especially considering that the method is simply using clustering to do label propagation.
> >
> > Based on the comparison of DON+Hard (and DON+Ground truth shown in the newly added supplementary material: Fig. 1-Supplementary) to DON+Soft, we can see that indeed the softness of the labels does improve the generalization ability of the model. However, this method still proves a significant improvement over the previous work that could allow scalability to learning novel objects in the wild, while the previous method is limited to a single object.
> >
> > > See RTPV's point about "across or between different scenes is critical"
> >
> > This has been fixed as suggested by RTPV.
> >
> > >One might argue that it would not be hard to extend the existing DON collection protocol (e.g. a camera orbiting a fixed object for which you know the class) by tossing in some distractor objects to make the data multi-view without additional labeling cost.
> >
> > Note that we assume the class is not known during data collection; we simply use it for evaluation purposes. We imagine a scenario in which the data protocol is extended in a different direction, i.e. the model is adapted with unsupervised scene segmentation to automatically segment multiple objects in its view, and learn their descriptors with appropriate augmentations. We do not intend to use any knowledge of the observed object's ground-truth class in any of the scenarios as that would limit the ability of the method to scale and learn novel objects in the wild.
> >
> > >Some relevant work the authors may want to look at:...
> >
> > We have added these works to the revision as per the Meta-Reviewer suggestion.

---

> > > ### Comment · Area_Chair_LDZa · 2021-08-25
> > > **Thanks - follow-up question regarding clutter**
> > >
> > > Thank you for the detailed explanation and for clearing up my misunderstanding. I need some additional time to process the remainder of the reply, but in the meantime I have a clarification question:
> > >
> > > On line 207-209, you state
> > > >To test this, we create new images, with unseen object instances in visually cluttered scenes, either
> > > through image collages or real photos (see Fig. 6). We manually label a selection of points on a
> > > set of 40 training images and the set of 20 test images, each having labelled semantically significant
> > > locations (when not occluded).
> > >
> > > Just  to confirm my understanding, the "unseen object instances in visually cluttered scenes" refers to the test set only, and the 40 training images mentioned above are not cluttered? In that case, the normalized error CDFs in Figure 7 are an aggregate over training and testing images?

---

> > > > ### Author Response · Authors · 2021-08-25
> > > > **Response**
> > > >
> > > > You are correct -  "unseen object instances in visually cluttered scenes" refers to the test set objects since clutter contains multiple objects, including more than half containing at least one of each of the object categories trained (e.g. both a hat and a shoe), see examples in Figure 6.
> > > >
> > > > The images dubbed `training set images` are images taken out of the dataset used for training, but these are only used for evaluation, similarly to the Sec 5.1. single object experiments in Florence et al. (we will make this clearer). Thus they contain objects of known instances but unseen images.
> > > >
> > > > For the multi-object test, as we state in L212: `The closest descriptors of the labelled points in the training images are found in query test set images.` - the training images are used as source points while queries are made in the test set images. Since each image has multiple points that are manually labelled (we give example with `... this includes semantically distinctive points such as tip, top and back of a hat`) depending on which of these are observable, we know the ground truth semantic correspondences between them (the query test images) and the source images (i.e. the labelled single object ones). As in our comment to Reviewer ZCDs `We imagined a scenario in which the robot recalls objects that it has already learned, and uses that memory to find corresponding points on novel objects at inference time`. A visual illustration of this is shown in Figure 8.
> > > >
> > > > For single object match CDF which we added for completeness, both the query and the source come from these single-object images.

---

### Decision · Program_Chairs · 2021-09-13

**Decision:**

Accept (Poster)

**Comment:**

This paper extends DON by making the object labels self-supervised and found via cluster assignment in pretrained visual feature space. This removes the prior limitation of DONs where the object-in-view needed to be labeled. The authors also show that augmenting the training set with cluttered images also improves generalization to multi-object scenes.

This paper currently has 2 weak rejects and 1 strong accept, with reviewer RTPV willing to go to bat for this paper.

*Strengths*: Reviewers appreciate the focus on scalable data for representation learning. Figure 6 represents an improvement over DON for multi-object matches, real robot results

*Weaknesses*: Reviewers 2SMq, RTPV say that the motivations and technical implementation details of Similarity Graphs (Sec 4.3.2) are unclear. 2SMq and ZCDs also recommend that the overall method be improved for clarity. Both 2SMq and ZCDs take issue with the test scenes only involve two objects classes and only contain one instance per class. I think this is a valid criticism, because it is quite possible that a cluster-based assignment for auto-labeling data may scale poorly as the number of classes is increased.

I have some questions about Figure 6 that I would like the authors to respond to in their rebuttal, and potentially RTPV as well.

It seems to me from reading the results in Figure 6 that the improved generalization to multi-object scenes is not a definitive property of the method, but rather due to the use of incorporating multi-object scenes into the training data (L208), and the use of soft labels. The benefit of training on multi-object scenes is obvious, so the remaining question is whether the proposed method contributes anything new to better descriptors in multi-object scenes beyond the data. Figure 6 seems to suggest this, but I noticed that the DON baseline performs about as well as the hard variant of the method on multi-object matches (hard to really understand the relative difference between blue/orange in Fig 6). This leads me to suspect that the improved performance of the soft labels is not due to the clustered label assignments, but rather the fact that the labels are just soft. In other words, a null hypothesis is that "one does not require a self-supervised method to recover soft labels; one could imagine a simple label smoothing trick applied to the hard labels would work as well as the DON + Soft Labels".

I would appreciate if the authors could check this hypothesis. Another experiment  to check the null hypothesis would be to train on self-supervised labels from single-object scenes only, and then test on multi-object scenes. If my suspicions are correct, both DON and DON + self-supervised labels will do equally poorly.

If the null hypothesis that "soft labels improve multi-object generalization for DON" is a property of the label softness and not the self-supervised clustering method itself, I think that does diminish the novelty of the paper, especially considering that the method is simply using clustering to do label propagation. Many prior works such as Caron et al. '18 have utilized this trick to provide labels for some other downstream learning task. Connecting unsupervised label propagation to a relatively new descriptor learning method is not guaranteed to "just work", so I appreciate the research being done here, but then I think some more compelling evidence around the promised scalability of the method needs to be shown. See RTPV's point about "across or between different scenes is critical" or handling more than 2 object classes as the other reviewers have pointed out. One might argue that it would not be hard to extend the existing DON collection protocol (e.g. a camera orbiting a fixed object for which you know the class) by tossing in some distractor objects to make the data multi-view without additional labeling cost.

Some relevant work the authors may want to look at:
- There have been some prior works like Grasp2Vec (Jang and Devin et al. 2018) that show dense, self-supervised object features learned in multi-object scenes (though not via pixel-to-pixel correspondences) and Object Contrastive Nets (Pirk et al. 2019) which also propose self-supervised object correspondences in multi-object scenes. Pirk et al. uses a single scene to learn object representations, but conceptually may not be limited by that.


---- update 20210904

I thank the authors for their detailed rebuttal and for the reviewers in engaging in discussion. One of the reviewers have updated their rating to weak accept, and given that the authors have addressed all my concerns as well, I will recommend an accept.